# Evaluation of antibody kinetics and durability in healthy individuals vaccinated with inactivated COVID-19 vaccine (CoronaVac): A cross-sectional and cohort study in Zhejiang, China

**Hangjie Zhang[1†], Qianhui Hua[2†], Nani Nani Xu[3], Xinpei Zhang[4], Bo Chen[5], Xijun Ma[6], Jie Hu[7], Zhongbing Chen[8], Pengfei Yu[7], Huijun Lei[8], Shenyu Wang[1], Linling Ding[1], Jian Fu[1], Yuting Liao[9], Juan Yang[9], Jianmin Jiang[1]\*, Huakun Lv[1]\***

[1]Department of Immunization Program, Zhejiang Provincial Center for Disease Control and Prevention, Hangzhou, China; [2]School of Medicine, Ningbo University, Ningbo, China; [3]Xihu District Center for Disease Control and Prevention, Hangzhou, China; [4]Shangyu District Center for Disease Control and Prevention, Shaoxing, China; [5]Kaihua District Center for Disease Control and Prevention, Quzhou, China; [6]Yuecheng District Center for Disease Control and Prevention, Shaoxing, China; [7]Department of Immunization Program, Jiaxing Center for Disease Control and Prevention, Jiaxing, China; [8]Longyou District Center for Disease Control and Prevention, Quzhou, China; [9]School of Public Health, Xiamen University, Xiamen, China

**\*For correspondence:**
jmjiang@cdc.zj.cn (JJ);
hklv@cdc.zj.cn (HL)

[†]These authors contributed equally to this work

**Competing interest:** The authors declare that no competing interests exist.

## Abstract

**Background:** Although inactivated COVID-19 vaccines are proven to be safe and effective in the general population, the dynamic response and duration of antibodies after vaccination in the real world should be further assessed.

**Methods:** We enrolled 1067 volunteers who had been vaccinated with one or two doses of CoronaVac in Zhejiang Province, China. Another 90 healthy adults without previous vaccinations were recruited and vaccinated with three doses of CoronaVac, 28 days and 6 months apart. Serum samples were collected from multiple timepoints and analyzed for specific IgM/IgG and neutralizing antibodies (NAbs) for immunogenicity evaluation. Antibody responses to the Delta and Omicron variants were measured by pseudovirus-based neutralization tests.

**Results:** Our results revealed that binding antibody IgM peaked 14–28 days after one dose of CoronaVac, while IgG and NAbs peaked approximately 1 month after the second dose then declined slightly over time. Antibody responses had waned by month 6 after vaccination and became undetectable in the majority of individuals at 12 months. Levels of NAbs to live SARS-CoV-2 were correlated with anti-SARS-CoV-2 IgG and NAbs to pseudovirus, but not IgM. Homologous booster around 6 months after primary vaccination activated anamnestic immunity and raised NAbs 25.5-fold. The neutralized fraction subsequently rose to 36.0% for Delta (p=0.03) and 4.3% for Omicron (p=0.004), and the response rate for Omicron rose from 7.9% (7/89)–17.8% (16/90).

**Conclusions:** Two doses of CoronaVac vaccine resulted in limited protection over a short duration. The inactivated vaccine booster can reverse the decrease of antibody levels to prime strain, but it does not elicit potent neutralization against Omicron; therefore, the optimization of booster procedures is vital.

**Funding:** Key Research and Development Program of Zhejiang Province; Key Program of Health Commission of Zhejiang Province/ Science Foundation of National Health Commission; Major Program of Zhejiang Municipal Natural Science Foundation; Explorer Program of Zhejiang Municipal Natural Science Foundation.

## Editor's evaluation

This study presents important evidence that boosting with the Sinovac Coronavac inactivated vaccine would provide considerable protection from ancestral SARS-CoV-2 in terms of elicited neutralizing antibodies but would offer minimal protection against Omicron subvariants. The evidence supporting the claims of the authors is solid, although using a dilution series instead of one plasma dilution for Omicron neutralization would have strengthened the study. The work will be of very wide interest to the biomedical community and beyond, since it points to the need for a better booster vaccine in China.

## Introduction

The coronavirus disease 2019 (COVID-19), a global health emergency caused by the severe acute respiratory syndrome coronavirus 2 (SARS-CoV-2), has led to unprecedented global healthcare and economic burdens (*Clark et al., 2020*). COVID-19 vaccines are indispensable for mitigating this situation and containing the ongoing pandemic, as shown by the decline in new and hospitalized COVID-19 cases since mass vaccination began (*Rossman et al., 2021*).

Inactivated COVID-19 vaccines, such as CoronaVac and BBIBP-CorV, were proven to be generally safe and effective in adults in several clinical trials (*Xia et al., 2022*; *Xia et al., 2020*; *Wu et al., 2021*) and are widely used in China and abroad. Nevertheless, basic questions remain about the vaccine-induced longevity of immunity in the population and the rate of breakthrough infections (*Kim et al., 2021*). Several studies have gathered immunogenicity data on antibody kinetics after vaccination and showed that neutralizing titers induced by two doses of inactivated vaccine peaked in month 2 and declined to 33.89% by month 6 (*Cheng et al., 2022*). However, it is important to provide more data on the enhancement and attenuation of immunological protection after vaccination in real-world studies.

The SARS-CoV-2 variants that have been classified as variants of interest or variants of concern (VOC) by the World Health Organization (WHO) are responsible for multiple waves of infection (*Dyson et al., 2021*) and the increased concerns about the protection provided by current vaccines (*Mlcochova et al., 2021*; *Altmann et al., 2021*). The Omicron (B.1.1.529) variant recently identified in South Africa has spread globally (*Karim and Karim, 2021*; *Viana et al., 2022*), raising concerns about the effectiveness of antibody therapies and vaccines to variants with multiple mutations (*Flemming, 2022*; *VanBlargan et al., 2022*; *Suzuki et al., 2022*; *Meng et al., 2022*). A recent real-world study in Israel suggested that a third dose of BNT162b2 vaccine was highly effective in preventing infection, severe disease, hospitalization, and death (*Barda et al., 2021*). Booster vaccines reinstate waning immunological memory and expand the breadth of immune responses to SARS-CoV-2 variants (*Goldberg et al., 2021*; *Zhu et al., 2022*; *Chen et al., 2022*; *Cao et al., 2021*; *Planas et al., 2022*). Therefore, the provision of booster vaccinations for SARS-CoV-2 is recommended by the WHO and is being implemented for fully vaccinated recipients in China and other countries. However, data are needed on the protective immune responses elicited by the boosters against VOC in mass vaccination campaigns.

We explored the dynamic responses and durations of antibodies against SARS-CoV-2 in individuals within 1 year of being vaccinated with an inactivated COVID-19 vaccine and speculated on the protection provided based on the attenuation of neutralizing antibody levels. Furthermore, we evaluated the presence of neutralizing antibodies against Delta and Omicron in volunteers boosted with a third dose of inactivated vaccine.

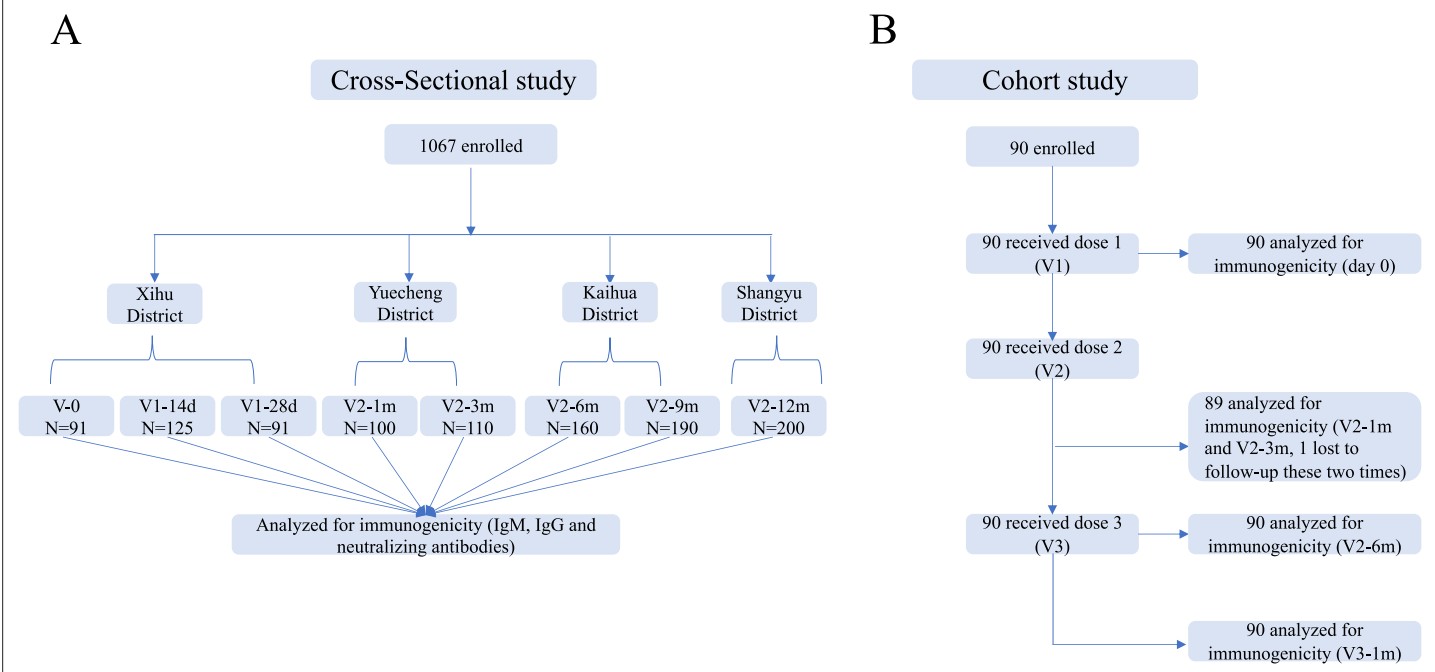

**Figure 1.** Schedule of sample collection. (**A**) Cross-sectional survey: a total of 1067 participants aged 18–59 were enrolled in five counties in Zhejiang, China. The participants had no previous vaccination or were vaccinated with one or two doses of CoronaVac. Venous blood (3–5 ml) was collected on day 0 (V-0, no vaccination), day 14±2 (V1-14d), and day 28±3 (V1-28d) after the first dose, and day 30±3 (V2-1m), day 90±7 (V2-3m), day 180±14 (V2-6m), day 270±14 (V2-9m), and day 365±30 (V2-12m) after the second dose. (**B**) Prospective cohort study: 90 healthy adults aged 18–80 years in Jiaxing city were recruited and administered 4 μg/0.5 mL of CoronaVac following a 3-shot vaccine schedule 28 days and 6 months apart. Following that, venous blood was collected from recipients at five timepoints: day 0 (Pre-V, before vaccination), day 30±3 (V2-1m), day 90±7 (V2-3m), and day 180±14 (V2-6m) after the second dose, and day 30±3 (V3-1m) after the third dose.

## Materials and methods
### Study design and participants

The cross-sectional investigation was conducted in five counties of Zhejiang Province, mainland China (Xihu, Yuecheng, Shangyu, Kaihua, and Longyou Districts), after nationwide COVID-19 vaccinations from May to October 2021. Potential participants aged 18–59 years who had no prior vaccinations or were vaccinated with one or two doses of CoronaVac (Sinovac Life Sciences, Beijing, China) were recruited from the community. Individuals with a history of infection with SARS-CoV-2 (based on epidemic surveillance system) or the use of blood products or immunosuppressive drugs were excluded. We randomly enrolled 1067 volunteers, including those on day 0 (V-0, no vaccination), day 14±2 (V1-14d), and day 28±3 (V1-28d) after the first vaccine dose, and day 30±3 (V2-1m), day 90±7 (V2-3m), day 180±14 (V2–6 m), day 270±14 (V2-9m), and day 365±30 (V2-12m) after the second dose and collected their venous blood samples (3–5 ml) to detect serum antibody levels (*Figure 1A*). This was not a longitudinal survey, as different subjects were enrolled at each point in time. We employed a questionnaire survey at blood drawing visits to gather demographic information.

In the prospective cohort study, we recruited 90 healthy adults aged 18–80 years from Jiaxing city, Zhejiang, in June 2021. The main exclusion criteria included previous or later SARS-CoV-2 infection; allergy to any ingredient included in the vaccine; those who had received any blood products or any research medicines or vaccines in the past month; those who had uncontrolled epilepsy or other serious neurological diseases, acute febrile disease, acute onset of chronic diseases, or uncontrolled severe chronic diseases; and those who were unable to comply with the study schedule. Subjects were administered 4 μg/0.5 mL of CoronaVac following a 3-shot vaccine schedule 28 days and 6 months apart. Following that, venous blood (3–5 ml) was collected from recipients at five time points: day 0 (Pre-V, before vaccination), day 30±3 (V2-1m), day 90±7 (V2-3m), and day 180±14 (V2-6m) after the second dose, and day 30±3 (V3-1m) after the third dose (*Figure 1B*).

## SARS-CoV-2-specific IgG and IgM assay

The commercial detection kit iFlash-2019-nCoV NAb assay (Shenzhen YHLO Biotech Co. Ltd., Shenzhen, China) was employed to measure the levels of IgG and IgM against SARS-CoV-2 spike glycoprotein (S) and nucleocapsid protein (N) by chemiluminescence immunoassay. Briefly, serum samples were allowed to form a complex with SARS-CoV-2 S- and N-protein antigen-coated paramagnetic microparticles, then an acridinium-ester-labeled ACE2 conjugate was added to competitively combine with the particles, forming another reaction mixture. The analyzer converted relative light units (RLUs) into an antibody titer (AU/mL) through a two-point calibration curve. An inverse relationship existed between the amount of SARS-CoV-2 NAb in the sample and the RLUs detected by the iFlash optical system. According to the manufacturer, titers of ≥10.0 AU/mL and ≥1.0 AU/mL are considered positive (or reactive) for IgG and IgM, respectively. IgG and IgM against SARS-CoV-2 receptor binding domain (RBD) were detected using a commercial ELISA kit (Bioscience Biotech Co. Ltd., Chongqing, China). The positive cutoff values for RBD-specific IgG and IgM antibodies were defined as titers of ≥1.0 AU/mL. All tests were performed according to the manufacturer's protocols (*Chan et al., 2021*; *Li et al., 2021*).

## Live virus neutralization antibody assays

The levels of neutralizing antibodies to live SARS-CoV-2 were assessed by the reduction in the cytopathic effect (CPE) in Vero cells with infectious SARS-CoV-2 strain 19nCoVCDC-Tan-HB01 (HB01) in a BSL-3 laboratory (*Zhang et al., 2021*). Briefly, serum samples were heat-inactivated for 30 min at 56 °C and successively diluted from 1:4 to the required concentration in a twofold series. An equal volume of challenge solution containing 100 TCID50 virus was added. After neutralization in a 37 °C incubator for 2 h, a $1.5–2.5×10^5$ /ml cell suspension was added to the wells. The CPE (cytopathic effect) on VeroE6 cells was analyzed at 4 days post-infection. NT50 (50% neutralization titer, the reciprocal of the highest dilution protecting 50% of the cells from virus challenge) was used to show the neutralization titers. NT50 above 1:4 was defined as positive.

## Pseudovirus-based neutralization test

Serum samples were also quantified for their content of SARS-CoV-2-neutralizing antibodies to wild-type (Wuhan), Delta (B.1.617.2), and Omicron (B.1.1.529) using the pseudovirus-based virus neutralization test (*Nie et al., 2020*). Briefly, serum samples and a positive or negative reference sample were each diluted 50 times with phosphate-buffered saline combined with 50 µl of pseudovirus diluent per well in a 96-well plate. The mixed sample/pseudovirus was incubated at 37 °C and 5% $CO_2$ for 1 hr. A $2×10^5$ /ml BHK-21-ACE2 cell suspension was added to each well of the plate containing the sample/pseudovirus mixture, then the plate was incubated in a 37 °C and 5% $CO_2$ cell incubator for 48 hr. Finally, the number of green-fluorescence-protein-positive cells per well was read with a porous plate imager (Tecan, Shanghai, SparkCyto). The results were determined by comparing the neutralized fraction using the following calculation: (1 – (fluorescence value of each well/average virus control value))×100% (*Karaba et al., 2022*). At least four wells were left blank for calibration to 0% inhibition.

## Statistical analysis

Sex, age, BMI, and other clinical characteristics were collected for each vaccination recipient. We used the medians and interquartile ranges (IQR) for age, and numbers (percentages) for categorical variables. Specific binding antibodies against SARS-CoV-2 (IgG, IgM) and neutralized fraction of SARS-CoV-2-neutralizing antibodies are presented as mean ± standard error (SEM). Neutralizing antibodies are presented as geometric mean titers (GMT), and their 95% confidence interval (CI) was calculated with Student's t distribution on log-transformed data and then back-transformed. Comparisons of titer-level differences between the two groups were performed using the paired Student's t-test. One-way analysis of variance (one-way ANOVA) was used to analyze the differences between the mean values at different timepoints. Correlations between NAb titers, neutralized fraction, and IgG/IgM levels were evaluated by Pearson's correlation coefficient. Statistical tests were two-sided, and we considered p-values of less than 0.05 as statistically significant. All statistical analyses were conducted in SPSS 18.0 (IBM Corporation, Armonk, NY, USA) and GraphPad Prism 9 (San Diego CA, USA).

**Table 1.** Baseline characteristics and anti-S/N antibody levels in the cross-sectional study.

| | V-0<br>N=91 | V1-14d<br>N=125 | V1-28d<br>N=91 | V2-1m<br>N=100 | V2-3m<br>N=110 | V2-6m<br>N=160 | V2-9m<br>N=190 | V2-12m<br>N=200 |
|---|---|---|---|---|---|---|---|---|
| Median age (IQR), years | 38(31,47) | 39(34,47) | 38(31,47) | 40(32,50) | 41(33,55) | 41(31,48) | 41(31,48) | 41(34,49) |
| **Sex** | | | | | | | | |
| Male | 37 | 45 | 37 | 49 | 54 | 80 | 85 | 75 |
| Female | 54 | 80 | 54 | 51 | 56 | 80 | 105 | 125 |
| **IgM** | | | | | | | | |
| Concentration (AU/ml) | 0.4±0.02 | 5.1±1.0 | 5.1±0.9 | 1.4±0.2 | 0.4±0.1 | 0.3±0.04 | 0.3±0.04 | 0.2±0.03 |
| Seropositivity (%) | 3.3 | 57.6 | 75.8 | 33.0 | 3.6 | 4.4 | 4.7 | 2.0 |
| **IgG** | | | | | | | | |
| Concentration (AU/ml) | 0.6±0.1 | 3.7±0.5 | 64.3±5.8 | 79.7±5.7 | 29.4±2.4 | 10.5±0.9 | 8.9±1.0 | 6.8±0.9 |
| Seropositivity (%) | 0.0 | 7.2 | 97.8 | 97.0 | 88.2 | 32.5 | 22.1 | 13.5 |

Data are n (%) or median (IQR), or mean ± SEM. The seropositivity rate is when positive concentration of anti-S/N antibody is 10.0 AU/mL (IgG) and ≥1.0 AU/mL (IgM) or more.

The online version of this article includes the following source data for table 1:

**Source data 1.** Baseline characteristics and anti-S/N antibody levels in the cross-sectional study.

## Results

### Study participant characteristics

We conducted a cross-sectional survey and recruited 1067 volunteers who had no vaccination or were vaccinated with one or two doses of CoronaVac in October 2020 or later in this multicenter study. Participants ranged in age from 18 to 59 years, with a median age of 40 years (IQR, [32-48]), and there was a balanced distribution of males (43.3%) and females (56.7%). Samples were collected at eight timepoints, including V-0 (n=91), V1-14d (n=125), and V1-28d (n=91) after vaccination with the first dose and V2-1m (n=100), V2-3m (n=110), V2-6m (n=160), V2-9m (n=190), and V2-12m (n=200) after vaccination with the second dose (*Figure 1A*). Demographic data for the vaccine recipients are summarized in *Table 1*.

In the prospective cohort study, we recruited 90 healthy adults who met all inclusion criteria and no exclusion criteria, including 40 (44.4%) males and 50 (56.6%) females with a median age of 64 years (IQR, [39-70]), 33.3% of whom had a BMI of ≥24.0 kg/m², and 33.3% had ≥1 underlying comorbidity (most commonly hypertension and diabetes) (*Table 2*). The participants were administered a standard dose of the CoronaVac vaccine on days 0 and 28 and a booster dose after month 7. Blood samples were collected at study visit 0 (Pre-V) before vaccination; visit 1 (V2-1m), visit 2 (V2-3m), and visit 3 (V2-6m) after vaccination with the second dose; and visit 4 (V3-1m) after the third dose (*Figure 1B*). None of participants had a history of laboratory-confirmed SARS-CoV-2 infection.

### Dynamics of antibody responses to primary vaccination

To explore the dynamic changes in humoral immune responses to the inactivated COVID-19 vaccine, we first evaluated the recipients' anti-S/N IgM and IgG development at different timepoints (*Figure 2A*). The titer of anti-S/N-IgM on day 0 increased to 5.1±1.0 AU/ml on day 28 after the first dose, though the seropositivity rate was 57.6%. The seropositivity rates of anti-S/N-IgM reached a peak of 75.8% (5.1±0.9 AU/ml) approximately 28 days after the first dose, while the seropositivity of anti-S/N-IgG reached 97.0% (79.7±5.7 AU/ml) approximately 28 days after the second dose. The titer of anti-S/N-IgM rapidly declined to 1.4±0.2 AU/ml, which is close to the threshold value, 28 days after the second dose, while anti-S/N-IgG declined to 10.5±0.9 AU/ml during month 6 after the second

**Table 2.** Baseline characteristics for the prospective cohort.

| | N=90 | P (%) |
|---|---|---|
| **Age group (years)** | | |
| 18–44 | 28 | 31.1 |
| 45–64 | 21 | 23.3 |
| 65–80 | 41 | 45.6 |
| **Sex** | | |
| Male | 40 | 44.4 |
| Female | 50 | 55.6 |
| **BMI (kg/m2)** | | |
| <18.5 | 3 | 3.3 |
| 18.5–23.9 | 57 | 63.4 |
| ≥24 | 30 | 33.3 |
| **Chronic conditions** | | |
| Yes | 30 | 33.3 |
| No | 60 | 66.7 |

The online version of this article includes the following source data for table 2:

**Source data 1.** Baseline characteristics for the prospective cohort.

dose. A small percentage of the population still had anti-S/N IgG antibodies, with seropositivity rates of 22.1% and 13.5%, respectively, during months 9 and 12 after the second dose (*Table 1*).

S protein RBD binding to the angiotensin converting enzyme 2 (ACE2) receptor is a critical initial step in the entry of SARS-CoV-2 into target cells (*Zuo et al., 2022*). We detected the anti-RBD IgM and IgG levels in the serum samples at several timepoints after the second dose (*Figure 2B*). The results were similar to those for anti-S/N antibodies, showing peak levels of anti-RBD-IgM (1.8±0.4 AU/ml) and anti-RBD-IgG (18.0±1.6 AU/ml) 1 month after the second dose, after which, the levels of both gradually waned. Furthermore, analysis showed a good correlation between IgM or IgG and anti-S/N and anti-RBD antibodies ($R^2$=0.7364, p<0.001; $R^2$=0.7170, p<0.001, *Figure 2C*).

Participants were tested with a live virus-based and pseudovirus-based neutralization assay. As depicted in *Figure 3A*, samples were negative for NAbs at the pre-vaccine baseline, and 87.0% of recipients had a NAb titer greater than 1:4 after the administration of the second dose, along with a GMT of 20.2 (95% CI: 16.3–24.9). Despite the decline observed in month 1, the values did not differ significantly between month 1 and month 3 (p=0.052). However, 12 months after immunization, 35.5% of the population were NAb-positive, with a GMT of 4.1 (95% CI: 3.7–4.5). Additionally, the neutralized fraction for the pseudovirus was significantly elevated (76.2%±1.6, p<0.001) at 1 month after the second dose and decreased slightly thereafter (*Figure 3B*). Correlation analysis showed poor correlation between NAb titers and anti-S/N IgM ($R^2$=0.014, p=0.243), NAb titers and anti-RBD IgM ($R^2$=0.010, p=0.322), NAb titers and anti-S/N IgG ($R^2$=0.087, p=0.003), neutralized fraction and anti-S/N IgM ($R^2$=0.084, p=0.003), neutralized fraction and anti-RBD IgM ($R^2$=0.048, p=0.028); whereas moderate correlations between NAb titers and anti-RBD IgG ($R^2$=0.121, p<0.001), NAb titer and neutralized fraction ($R^2$=0.135, p<0.001), neutralized fraction and anti-S/N IgG ($R^2$=0.539, p<0.001), neutralized fraction and anti-RBD IgG ($R^2$=0.471, p<0.001).

## Antibody responses before and after booster immunization

With the dampening of antibody responses to the CoronaVac vaccine, we gave the cohort of 90 individuals booster immunizations up to 6 months after the second dose, and the antibody-titer-level distributions are shown in the violin plot in *Figure 4*. At month 1 after the second dose, the seropositivity of anti-S/N and anti-RBD IgGs were 96.7% and 100.0% and reached peak levels of 67.4±5.0 AU/ml and 9.4±0.8 AU/ml, respectively. After which point, they slowly diminished over time to 9.4±1.6

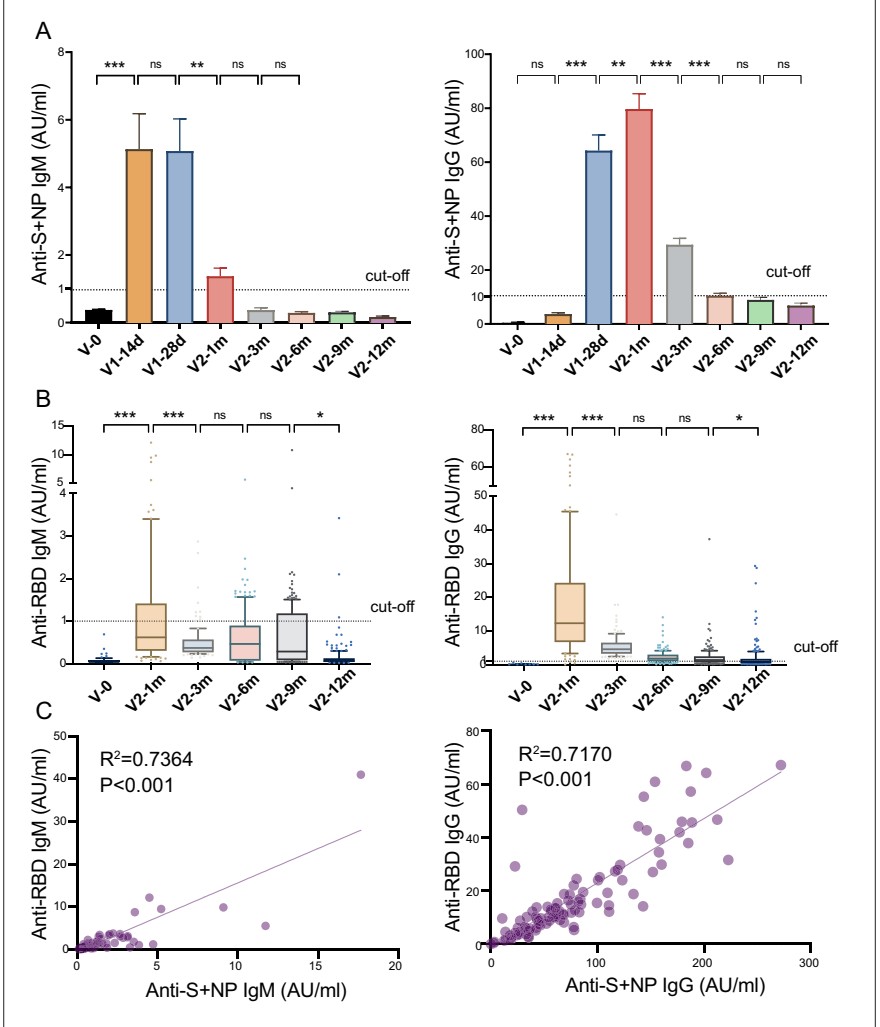

**Figure 2.** Anti-SARS-CoV-2-specific IgG and IgM levels induced by inactivated COVID-19 vaccines. (**A, B**) Dynamic changes in anti-S/ N- (**A**) and anti-RBD- (**B**) specific IgM/IgG in serum samples from CoronaVac-vaccinated participants at V-0, V1-14d, V1-28d, V2-1m, V2-3m, V2-6m, V2-9m, and V2-12m. (**C**) Correlation between levels of anti-S/anti-N- and anti-RBD-specific antibodies in IgM (left) or IgG (right) at V2-1m. Dates are presented as mean ± SEM. One-way analysis of variance was used for comparison. Correlations were assessed using Pearson's correlation coefficient. Two-tailed p values were calculated. ns, not significant, * $p<0.05$, **$p<0.01$, ***$p<0.001$.

The online version of this article includes the following source data for figure 2:

**Source data 1.** Anti-SARS-CoV-2-specific IgG and IgM levels induced by inactivated COVID-19 vaccines.

AU/ml and 2.8±0.2 AU/ml, respectively, in month 6. Injection of the booster dose stimulated these levels back to 131.3±8.6 AU/ml and 21.7±1.8 AU/ml at month 1 post-booster, 14.0- and 7.8-fold increases from the lowest point, respectively (*Figure 4A*).

After the primary two doses and the third booster dose, a similar increasing trend was observed in the two neutralization test results (*Figure 4B*). The GMT of the NAb titer peaked at 29.4 (95% CI: 23.5–24.2) and dropped to 6.6 (95% CI: 5.4–8.0) at month 6, which was a 4.5-fold attenuation of the peak value, with the total seropositivity dropping from 98.9% to 57.8%. After the booster dose, the GMT increased to 168.2 (95% CI: 139.7–202.6), that is, 25.5-fold higher, at month 6, which was 5.7-fold higher than the first peak. The result showed that the neutralized fraction for the pseudovirus peaked at 72.7% ± 1.6% before gradually decreasing to 21.4% ± 1.7% in month 6 but increased to 84.3% ± 1.7% after the booster.

The levels of antibodies grouped by age (i.e. 18–44 y.o., 45–64 y.o., and≥65y.o.), sex, BMI, and chronic conditions at each monitoring point are presented in *Table 3*. The Nab titer showed a

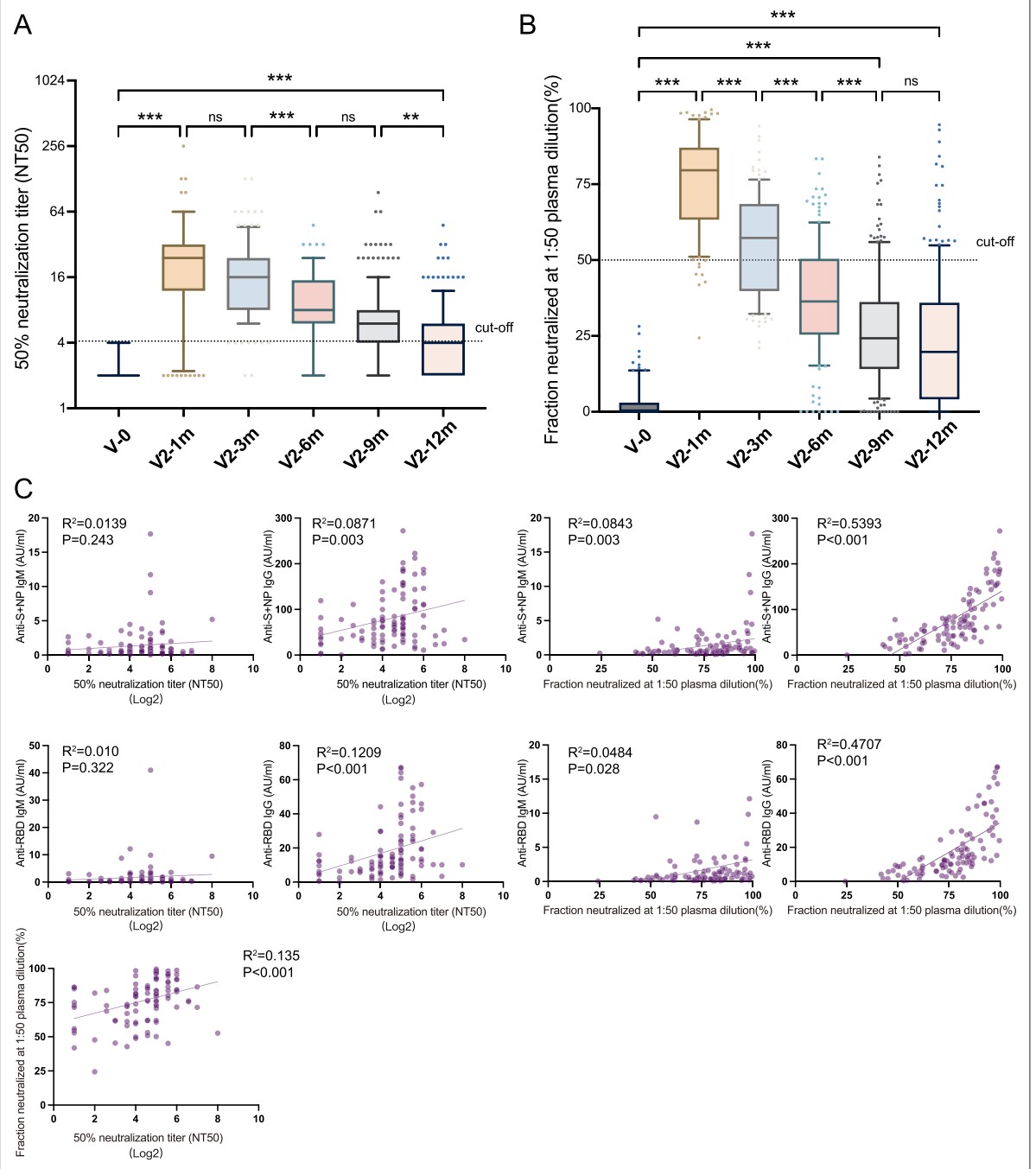

**Figure 3.** Neutralizing antibodies induced by inactivated COVID-19 vaccines. (**A, B**) Dynamic changes in GMT of NAb titer (**A**) and neutralized fraction (**B**) in serum samples from CoronaVac-vaccinated participants at V-0, V2-1m, V2-3m, V2-6m, V2-9m, and V2-12m. (**C**) Correlation among levels of anti-SARS-CoV-2-IgM and IgG, GMT of NAb titer, and neutralized fraction at V2-1m. One-way analysis of variance was used for comparison. Correlations were assessed using Pearson's correlation coefficient. Two-tailed p values were calculated. ns, not significant, * p<0.05, **p<0.01, ***p<0.001.

The online version of this article includes the following source data for figure 3:

**Source data 1.** Neutralizing antibodies induced by inactivated COVID-19 vaccines.

statistically significant difference among age groups during months 1–6 after the primary vaccination, but there was no significant difference (p=0.369) after the booster dose. Females exhibited a higher NAb titer than males [199.3 (95% CI: 159.2–249.5) vs 136.1 (95% CI: 99.9–185.4), p=0.039] after the booster vaccination. However, there were no statistically significant differences in antibody titer levels between the different BMI or chronic conditions groups.

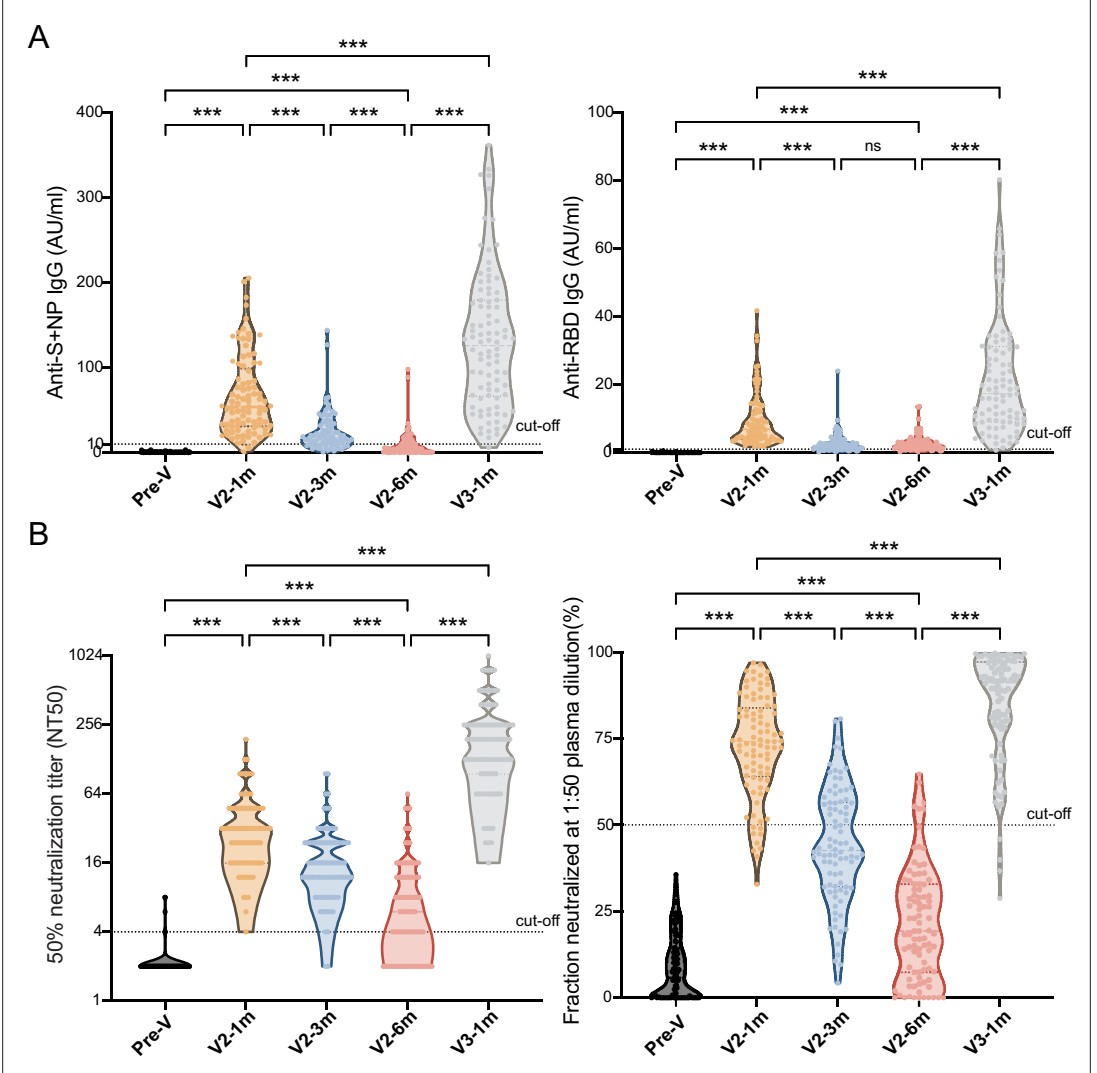

**Figure 4.** Comparisons of anti-SARS-CoV-2-specific IgG and IgM levels and neutralizing activity before and after booster immunization. (**A, B**) Dynamic changes in anti-S/N IgM and IgG (**A**), GMT of NAb titer, and neutralized fraction (**B**) in serum samples from CoronaVac-vaccinated participants at V-0, V2-1m, V2-3m, V2-6m, and V3-1m. One-way analysis of variance was used for comparison. Two-tailed P values were calculated. ns, not significant, * p< 0.05, **p < 0.01, ***p < 0.001.

The online version of this article includes the following source data for figure 4:

**Source data 1.** Comparisons of anti-SARS-CoV-2-specific IgG and IgM levels and neutralizing activity before and after booster immunization.

## Antibody responses to Delta and Omicron variants

Mutations in the RBD may lead to a reduction in the antibody neutralization susceptibility of VOC (*Liu et al., 2021*). We furthermore measured the levels of neutralizing antibodies against the Delta (B.1.617.2) and Omicron (B.1.1.529) variants from serum samples in month 1 after the primary and booster immunizations (*Figure 5*). Of note, in individuals vaccinated with two doses of the inactivated vaccine, the neutralized fraction for the pseudovirus against the Delta variant and, in particular, the Omicron variant were much lower compared with that against the Wuhan strain (28.4% vs 72.4%, 0.5% vs 72.4%, p≤0.001). However, the booster vaccination gave rise to a slight increase in neutralizing activity against the variants. The neutralized fraction subsequently rose to 36.0% for Delta (p=0.03) and 4.3% (p=0.004) for Omicron after booster dose of inactivated vaccine. The response rate (neutralized fraction >0%) for Omicron rose from 7.9% (7/89) in the primary two doses to 17.8% (16/90) after booster dose. Therefore, our results showed that the booster of Coronavac did not elicit potent neutralization against Omicron BA.1, although booster dose slightly increased antibody responses.

**Table 3.** The influence of age to GMT and Seropositivity.

| Time point | Statistic | Total | 18–44 years | 45–64 years | ≥65 years | p |
|---|---|---|---|---|---|---|
| Pre-V | GMT | 2.1 | 2.2 | 2.1 | 2.1 | 0.908 |
| | 95% CI | 2.0–2.2 | 1.9–2.4 | 1.9–2.4 | 2.0–2.3 | |
| | Seropositivity (%) | 3.3 | 3.6 | 4.8 | 2.4 | |
| V2-1m | GMT | 29.4 | 33.2 | 36.8 | 24.3 | 0.019 |
| | 95% CI | 23.5–24.2 | 25.3–43.4 | 24.3–55.7 | 20.0–29.5 | |
| | Seropositivity (%) | 98.9 | 100 | 95.2 | 100 | |
| V2-3m | GMT | 15.5 | 22.9 | 16.0 | 11.5 | <0.01 |
| | 95% CI | 13.3–18.0 | 17.0–30.9 | 12.5–20.5 | 9.4–14.2 | |
| | Seropositivity (%) | 95.5 | 100 | 95.2 | 92.7 | |
| V2-6m | GMT | 6.6 | 10.2 | 7.9 | 4.4 | <0.01 |
| | 95% CI | 5.4–8.0 | 7.0–14.7 | 5.1–12.3 | 3.5–5.7 | |
| | Seropositivity (%) | 57.8 | 82.1 | 66.7 | 35.6 | |
| V3-1m | GMT | 168.2 | 217.2 | 152.3 | 148.7 | 0.369 |
| | 95% CI | 139.7–202.6 | 165.7–287.7 | 97.3–238.5 | 110.7–199.7 | |
| | Seropositivity (%) | 100.0 | 100 | 100 | 100 | |

The seropositivity rate is when positive NT50 is above 1:4.
GMT = geometric mean titers.

## Discussion

Inactivated vaccines have been widely used worldwide, mainly to reduce the COVID-19 severity and hospitalization rate and number of related deaths (*Jara et al., 2021*; *Medeiros-Ribeiro et al., 2021*). However, there has been no correlation between immunization and protection or duration of protection demonstrated for inactivated COVID-19 vaccines. Our study of post-vaccination antibody kinetics showed that IgM levels peaked 14–28 days after one dose of CoronaVac, after which point, they declined rapidly. Two doses of the completely inactivated vaccine induced high levels of IgG and neutralizing antibodies, which peaked approximately 1 month after vaccination and declined slightly over time. The vaccine-induced immunity had notably waned to lower levels 6 months later and became undetectable in the majority of individuals 12 months later.

Previous studies have shown that antibody responses usually decline over time after an initial COVID-19 vaccination. Following vaccination with the Pfizer BNT162b2 vaccine, humoral responses were substantially decreased after 6 months among healthcare workers in Israel of 65 years of age or older, especially men (*Levin et al., 2021*). In addition, a significant trend of declining S-antibody levels was observed following AstraZeneca ChAdOx1 and BNT162b2 administration, with levels reducing by about 5-fold and 2-fold 21–41 days and 70 days or more after the second dose, respectively (*Shrotri et al., 2021*). It was reported that the immunity provided by inactivated COVID-19 vaccines, such as CoronaVac and Sinopharm/BBIBP, endured for 6 months, with a NAb GMT of 6.8 (95%CI: 5.2–8.8) and 2.31 (95%CI: 2.1–2.6) in month 6 after the two doses and seropositivity of 35% and 61%, respectively (*Cheng et al., 2022*; *Zeng et al., 2022*). In our study, the seropositivity and GMT of CoronaVac remained at 57.8% and 6.6 (95% CI: 5.4–8.0), respectively, for 6 months after the primary vaccination, which is comparable to prior reported data on inactivated vaccines (*Cheng et al., 2022*), although the immunogenicity was lower than that of other types of COVID-19 vaccines. However, the neutralization antibody response levels of different vaccines are difficult to directly compare because of a lack of standardized laboratory methods for SARS-CoV-2 neutralization and differences in experimental conditions (*Chen et al., 2021*).

We also assessed the dynamics of antibody levels in a small prospective longitudinal cohort 6 months following two vaccination doses and found they were consistent with the above findings.

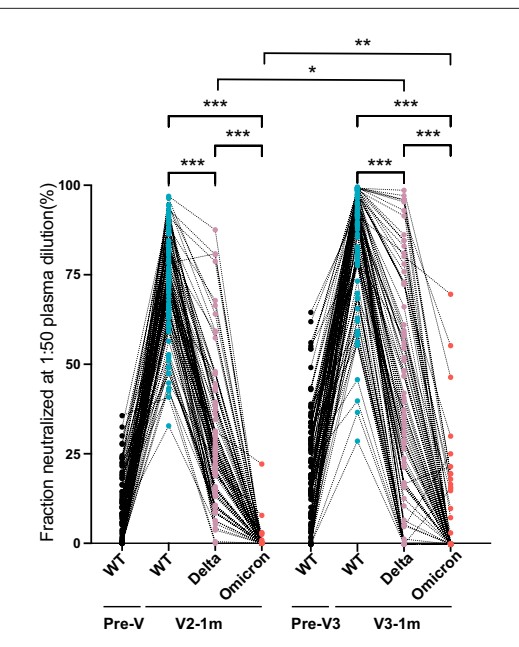

**Figure 5.** Antibody responses to Delta and Omicron variants. Neutralized fraction of Wuhan strain, Delta (B.1.617.2), and Omicron (B.1.1.529) variants for CoronaVac primary- and booster-vaccinated participants, as evaluated by pseudovirus-based neutralization test. The paired Student's t-test and one-way analysis of variance were used for comparison. Two-tailed p values were calculated. * p<0.05, **p<0.01, ***p<0.001.

The online version of this article includes the following source data for figure 5:

**Source data 1.** Antibody responses to Delta and Omicron variants.

Importantly, a booster injection around 6 months after the primary vaccination activated the anamnestic immunity and raised immune components (such as NAbs) to higher levels. Additionally, several factors, such as age group, sex, obesity, and chronic conditions affecting neutralizing antibody levels, were analyzed. We found that initial and booster-vaccine-elicited neutralizing antibody titers was weaker in older than younger adults, similar to the findings described in many other reports (*Meng et al., 2021*; *Wu et al., 2021*; *Liao et al., 2022*). Even so, Li, et al. showed that among Delta cases, the risk of ≥60 year age group developing pneumonia was 66% lower in the full vaccination age groups compared with no vaccination. And ≥60 year Delta cases the inactivated vaccine booster dose had 86% lower risk of developing pneumonia similar to 18–59 year cases (*Li et al., 2022*). There is evidence of vaccination-induced protection against severe COVID-19 in people with excess weight or obesity of a similar magnitude to that in people of a healthy weight (*Piernas et al., 2022*), and we found the COVID-19 neutralizing antibody titer was not associated with bodyweight. The comorbidities had no significant effect on NAb levels in our study, which agrees with another other study that demonstrated that inactivated COVID-19 vaccines had good immunogenicity and safety in older patients with hypertension and/or diabetes mellitus (*Zhang et al., 2022*). NAbs levels following the CoronaVac booster vaccine was found to have an association with sex, though the BNT162b2 mRNA vaccine was reported to induce higher antibody levels in women than men (*Lustig et al., 2021*).

The live SARS-CoV2 neutralizing antibody test has been used as a gold standard for evaluating vaccine immunogenicity in clinical trials and has shown correlations with protection from COVID-19 for real-world vaccinations (*Gilbert et al., 2022*; *Khoury et al., 2021*). In our study, we compared different methods of detecting the vaccine immunogenicity of serum samples with binding antibodies (IgM/IgG) and live and pseudovirus-based virus neutralizing antibodies. We found the correlations between both anti S/N and RBD IgM and IgG to be very strong. Moreover, the titers of NAbs to live SARS-CoV-2 were correlated with anti-SARS-CoV-2 IgG or NAbs to pseudovirus; however, NAbs to live SARS-CoV-2 did not correlate well with anti-SARS-CoV-2 IgM levels. This might be attributable to the rapid attenuation of IgM at the incipient stage and its further decrease over time and the contribution of IgG to increases in neutralizing antibody and IgM. Therefore, anti-SARS-CoV-2- IgG and pseudovirus-based neutralizing antibody detection methods can be applied to partly substitute for serological tests of live virus-neutralizing tests.

The Delta variant has eight mutations of the S protein, two of which are within the RBD, while there are 30 spike mutations in Omicron variant BA.1 (*Kumar et al., 2022*; *Zhao et al., 2022*). Our results revealed the capability of the SARS-CoV-2 Delta and Omicron variants escaped the antibodies induced by the inactivated vaccine after both primary and booster doses, especially for Omicron with a steep reduction in neutralization. Planas et al. demonstrated that Omicron was able to evade the recognition of most therapeutic monoclonal antibodies completely or partially. Sera from recipients of the Pfizer or AstraZeneca vaccine five months after complete vaccination or COVID-19-convalescent

patients collected 6 or 12 months after symptoms, showed low or no neutralizing activity against Omicron (*Planas et al., 2022*). Although Omicron had a high immune escape rate after inactivated vaccine administration and led to breakthrough infection with severe and fatal cases, a study of the University of Hong Kong revealed that three-dose inactivated vaccine could provide 98.1% protection of severe symptoms and death induced by Omicron in elderly people (>60 years old), and that of the BioNTech vaccine was 98.3% (*Zhou et al., 2023*; *Li et al., 2022*).

However, there were some limitations to this investigation. First, the sample size in this study was based on practical considerations rather than statistical power calculations, and the distribution of age and sex were not very well balanced in the study. Second, vaccine efficacy was not calculated, as we did not compare the NAb titers generated by vaccination with those of convalescent COVID-19 patients or a serum standard panel in parallel. Third, we assessed only serum antibody responses, and further evaluations focusing on memory and cellular immunity are needed. Forth, we detected the neutralization activity to Delta and Omicron with pseudovirus-based neutralizing test, only calculated the fraction neutralized at 1:50 plasma dilution (%), not the NT50.

In conclusion, our study suggests that vaccination against SARS-CoV-2 with two doses of CoronaVac can trigger humoral responses in the majority of vaccination recipients aged 18–59 years. Although binding and neutralizing antibodies were still detectable at 12 months post-vaccine, serum antibody levels tended to decrease over time, and Delta and Omicron variants may be able to more efficiently evade the antibodies induced by the inactivated vaccine with time. Despite the booster dose of inactivated vaccine can reverse the decrease of antibody levels to prime strain and heighten the cross immune response to Delta, it does not elicit potent neutralization against Omicron, which is circulating now. So, the optimization of the booster procedure such as heterologous boost immunization using viral-vector-, nucleic-acid-, and protein-based vaccines is necessary.

## Acknowledgements

We thank the staff from the local CDCs and the local designated hospitals for their help with the field survey. We also sincerely thank professor Xiaofeng Qin in Suzhou Institute of Systematic Medicine for assisting us to detect the pseudovirus-based neutralization antibody. This work was supported by the Key Research and Development Program of Zhejiang Province (2021C03200); the Key Program of Health Commission of Zhejiang Province/ Science Foundation of National Health Commission (WKJ-ZJ-2221); the Major Program of Zhejiang Municipal Natural Science Foundation (LD22H190001); the Explorer Program of Zhejiang Municipal Natural Science Foundation (LQ23H100001).

## Additional information

### Funding

| Funder | Grant reference number | Author |
| --- | --- | --- |
| Key Research and Development Program of Zhejiang Province | 2021C03200 | Hangjie Zhang<br>Nani Nani Xu<br>Bo Chen<br>Yuting Liao<br>Juan Yang<br>Jianmin Jiang<br>Huakun Lv |
| Key Program of Health Commission of Zhejiang Province/ Science Foundation of National Health Commission | WKJ-ZJ-2221 | Jianmin Jiang<br>Huakun Lv |
| Major program of Zhejiang Municipal Natural Science Foundation | LD22H190001 | Hangjie Zhang |
| Explorer Program of Zhejiang Municipal Natural Science Foundation | LQ23H00001 | Hangjie Zhang |

| Funder | Grant reference number | Author |
|---|---|---|

The funders had no role in study design, data collection and interpretation, or the decision to submit the work for publication.

## Author contributions

Hangjie Zhang, Software, Formal analysis, Writing - original draft; Qianhui Hua, Yuting Liao, Formal analysis; Nani Nani Xu, Xinpei Zhang, Bo Chen, Xijun Ma, Jie Hu, Zhongbing Chen, Pengfei Yu, Huijun Lei, Shenyu Wang, Resources; Linling Ding, Data curation, Formal analysis, Project administration; Jian Fu, Data curation, Project administration; Juan Yang, Formal analysis, Investigation; Jianmin Jiang, Huakun Lv, Funding acquisition, Writing - original draft, Project administration, Writing - review and editing

## Author ORCIDs

Hangjie Zhang (ORCID) http://orcid.org/0000-0001-7334-2536
Jianmin Jiang (ORCID) http://orcid.org/0000-0003-3563-4006
Huakun Lv (ORCID) http://orcid.org/0000-0002-3420-6942

## Ethics

We state that we conformed with the Helsinki Declaration of 1975 (as revised in 2008) concerning Human and Animal Rights, and that we followed out the policy concerning Informed Consent as shown on Springer.com. The study protocol and informed consent form were approved by the Medical Ethics Committee of The Zhejiang Center for Disease Control and Prevention (2021-044-01). Written informed consent was obtained from all study participants.

## Decision letter and Author response

Decision letter https://doi.org/10.7554/eLife.84056.sa1
Author response https://doi.org/10.7554/eLife.84056.sa2

# Additional files

## Supplementary files

• Supplementary file 1. Supplementary tables for supplemental instruction. Table S1 Baseline characteristics for the cross-sectional survey; Table S2 Correspondence of neutralized fraction (%) and serum dilution; Table S3 The influence of sex, BMI, and chronic condition to GMT and Seropositivity.

• Supplementary file 2. Statistical analysis.

• MDAR checklist

## Data availability

All data generated or analysed during this study are included in this published article and its supplementary information files; source data files have been provided on Dryad: https://doi.org/10.5061/dryad.ghx3ffbsw.

The following dataset was generated:

| Author(s) | Year | Dataset title | Dataset URL | Database and Identifier |
|---|---|---|---|---|
| Zhang H, Hua Q, NN Xu, Zhang X, Chen B, Ma X, Hu J, Chen Z, Yu P, Lei H, Wang S, Ding L, Fu J, Liao Y, Yang J, Jiang J, Lv H | 2023 | Data for: Evaluation of antibody kinetics and durability in health individuals vaccinated with inactivated COVID-19 vaccine (CoronaVac): a cross-sectional and cohort study in Zhejiang, China | https://doi.org/10.5061/dryad.ghx3ffbsw | Dryad Digital Repository, 10.5061/dryad.ghx3ffbsw |

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
