## [Editor Report]

This study presents important evidence that boosting with the Sinovac Coronavac inactivated vaccine would provide considerable protection from ancestral SARS-CoV-2 in terms of elicited neutralizing antibodies but would offer minimal protection against Omicron subvariants. The evidence supporting the claims of the authors is solid, although using a dilution series instead of one plasma dilution for Omicron neutralization would have strengthened the study. The work will be of very wide interest to the biomedical community and beyond, since it points to the need for a better booster vaccine in China.

---

## [Decision Letter]

**Decision letter after peer review:**

Thank you for submitting your article "Evaluation of antibody kinetics and durability in health subjects vaccinated with inactivated COVID-19 vaccine (CoronaVac): A cross-sectional and cohort study in Zhejiang, China" for consideration by *eLife*. Your article has been reviewed by 2 peer reviewers, including Alex Sigal as Reviewing Editor and Reviewer #1, and the evaluation has been overseen by Neil Ferguson as the Senior Editor.

Essential revisions:

Given the current shift in COVID-19 policy in China from Zero-Covid to minimal restrictions, a major infection wave is taking place. This paper shows that boosting with the Sinovac Coronavac inactivated vaccine would provide considerable protection from the ancestral virus and the no longer circulating Δ variant in terms of elicited neutralizing antibodies. But it would offer minimal protection against Omicron subvariants.

This data (Figure 5) points to the need for a better booster vaccine in China. After careful consideration, the Reviewers agreed that given the time-sensitive nature of the data, the paper can be re-submitted without further experimental work. The authors should ensure that the following points are addressed:

(1) The bottom line, that Coronavac does not elicit potent neutralization against Omicron, should be unequivocally stated.

(2) There are multiple issues with the presentation of the work as outlined by both Reviewers which should be fixed in the revised manuscript. As an example, the meaning of "neutralization inhibition rate" in Figure 5 is critical to understand the figure but is hard to decipher without carefully reading the Materials and methods. The relevant information (e.g. that testing was done at a 1:50 plasma dilution) should be accessible in the figure (e.g., change the axis label to something like "fraction neutralized at 1:50 plasma dilution"). A more standard ID50 would be better if the authors have the serial dilution data.

*Reviewer #1 (Recommendations for the authors):*

While the paper is an important one, the major weakness is the way the pseudovirus neutralization assays were done. In addition, whether any of the participants were previously infected is unclear.

(1) The pseudovirus neutralization assay is done only at the 1:50 concentration. This weakens the result and presents them in a way that is non-standard to the field. At least for the booster results, a standard dilution series should be performed and a dose-response curve fitted to obtain the ID50. The limit of detection should also be included, and be clearly presented for all figures.

(2) It would be very useful to know what proportion of study participants were previously infected and if there are differences between the previously infected and uninfected individuals. This is clearly complicated by the fact that the vaccine is inactivated virus and therefore discrimination by the presence of anti-nucleocapsid antibodies will not work. However, any clinical record of infection could be used. If none, this should be stated.

(3) Given the data, it is unclear how the authors can conclude on line 260 that: "protective capability of a primary vaccination series is enhanced following booster doses and optimization of the booster procedure" unless they are not talking about their study. The boost doesn't work as far as eliciting neutralizing antibodies against Omicron, which is what is circulating now, and this should be stated outright.

*Reviewer #2 (Recommendations for the authors):*

Line 115: 'critical' value. Did the authors mean threshold here?

Lines 135-139 are unclear in terms of the exact correlations being tested for.

Line 160 – may be better to use the term 'primary' course of vaccination.

Line 177: I would not use the word unprecedented. Rather could say they have been used widely.

Line 210-212: this does not make sense.

The references omit some of the key papers in the field. Where possible the authors should cite primary sources eg for the emergence of omicron Viana et al., Nature. For comparative biology of spike between omicron and δ, there are a number of possibilities including https://doi.org/10.1038/s41586-022-04462-1, https://www.tandfonline.com/doi/full/10.1080/22221751.2021.2023329, https://doi.org/10.1038/s41586-022-04474-x, and others. Some of the papers also report an increased breadth of neutralisation against omicron with booster doses that should be included in citations and compared to results observed in this study with boosting. https://doi.org/10.1038/s41586-021-04389-z.

In addition, the authors discuss the impact of age on response to vaccines but have not adequately cited papers on vaccine efficacy or in vitro neutralsiation / T cell responses in the elderly. This is clearly a key point for the Chinese population and needs further discussion, especially in light of recent developments in covid control policies.

The rationale for using an assay that detects N and S antibodies together without distinguishing them is unclear and should be addressed. In my view, this is a weakness.

Figure 4 should show data for Delta and Omicron. The data appear to be for WT only.

Table 1: Please define how the positive rate was determined.

Table 3: The 'positive rate' should be defined. Is it a GMT of >20 for example?

Table S2: please define P1 and P2 in the table legend. There should be 95% CI or sd around the inhibition rate %.

Table S3: The 'positive rate' should be defined.

[Editors’ note: further revisions were suggested prior to acceptance, as described below.]

Thank you for resubmitting your work entitled "Evaluation of antibody kinetics and durability in health subjects vaccinated with inactivated COVID-19 vaccine (CoronaVac): A cross-sectional and cohort study in Zhejiang, China" for further consideration by *eLife*. Your revised article has been evaluated by Neil Ferguson (Senior Editor) and a Reviewing Editor.

The manuscript has been improved but there are some remaining issues that need to be addressed, as outlined below:

The authors partially completed the referee corrections, and this paper does not require an additional round of review. However, the presentation of the work should be corrected before publication.

(1) The most significant result of the study – that the inactivated vaccine booster does not elicit potent neutralizing antibodies against Omicron, should be stated in the abstract.

(2) Title: "Evaluation of antibody kinetics and durability in health subjects…" should be changed to "Evaluation of antibody kinetics and durability in healthy…". Also, the term "subjects" is outdated, and the authors should consider using "individuals" or "participants".

(3) Figure 2: that cross-sectional and not cohort (if that is the case) study participants were analyzed should be stated in the legend. If the participants at each time point are not the same, there should be no connecting lines between points (e.g., a bar graph should be used).

(4) Figure 3A-B: labels on graph missing "2" (e.g., V-1m instead of V2-1m).

---

## [Author Response]

Essential revisions:(1) The bottom line, that Coronavac does not elicit potent neutralization against Omicron, should be unequivocally stated.

Thanks for your great suggestion. We have mortified the description in line 44-45, 347-350. Based on our data, we found Coronavac does not elicit potent neutralization against Omicron after the third dose as the editor' and reviewers' comments. Other studies draw similar conclusions. Li, et al. showed the neutralizing antibody GMTs against the Delta variant at day 14 after the homologous booster was 8.2 (95% CI=6.6, 10.1) with 56% seroconversion (Li, et al. 2022). Fanglei Zuo, et al. showed the 90% neutralization titer (NT90) value against Β, Delta and Omicron VOC was 5, 5 and 5 respectively after homologous booster with inactivated vaccine (Zuo, et al. 2022). However, we considered the method only used the fraction neutralized at 1:50 plasma dilution (%) with pseudovirus-based neutralizing test was less powerful, so we did not emphasize this result unequivocally.

(2) There are multiple issues with the presentation of the work as outlined by both Reviewers which should be fixed in the revised manuscript. As an example, the meaning of "neutralization inhibition rate" in Figure 5 is critical to understand the figure but is hard to decipher without carefully reading the Materials and methods. The relevant information (e.g. that testing was done at a 1:50 plasma dilution) should be accessible in the figure (e.g., change the axis label to something like "fraction neutralized at 1:50 plasma dilution"). A more standard ID50 would be better if the authors have the serial dilution data.

Thanks for your great suggestion on improving the accessibility of our manuscript. We have corrected the y-axis label in figures and figure legends. The multiple issues by both Reviewers we have fixed in the revised manuscript and answered the comments point by point. Due to the few reagents of pseudovirus of Delta and Omicron, we could not detect all samples with the serial dilution. We tried to detect two serum samples by serial dilution showed in Table S2 as a limited reference, but it cannot work because of the different calculated method and observed outcome. Karaba, et al. detected with the similar method (pseudoneutralization/ACE2 inhibition measurement) showed that a cutoff of 20% ACE2 inhibition is associated with measurable live virus–neutralizing antibody, including versus VOCs (Karaba, et al).

Reference

1. Li J, Hou L, Guo X, Jin P, Wu S, Zhu J, et al. Heterologous AD5-nCOV plus CoronaVac versus homologous CoronaVac vaccination: a randomized phase 4 trial. Nat Med. 2022;28(2):401-9.

2. Zuo F, Abolhassani H, Du L, Piralla A, Bertoglio F, de Campos-Mata L, et al. Heterologous immunization with inactivated vaccine followed by mRNA-booster elicits strong immunity against SARS-CoV-2 Omicron variant. Nat Commun. 2022;13(1):2670.

3. Karaba AH, Zhu X, Liang T, Wang KH, Rittenhouse AG, Akinde O, et al. A third dose of SARS-CoV-2 vaccine increases neutralizing antibodies against variants of concern in solid organ transplant recipients. Am J Transplant. 2022;22(4):1253-60.

Reviewer #1 (Recommendations for the authors):While the paper is an important one, the major weakness is the way the pseudovirus neutralization assays were done. In addition, whether any of the participants were previously infected is unclear.(1) The pseudovirus neutralization assay is done only at the 1:50 concentration. This weakens the result and presents them in a way that is non-standard to the field. At least for the booster results, a standard dilution series should be performed and a dose-response curve fitted to obtain the ID50. The limit of detection should also be included, and be clearly presented for all figures.

Thanks for your great suggestion. We agree it is a weaken for our study only to calculate the fraction neutralized at 1:50 plasma dilution (%) with pseudovirus-based neutralizing test, not the NT50. We have discussed this limitation in the discussion. We designed the study early in 2021, when there were not many SARS-CoV-2 mutant strains. The primary objective was to evaluate the antibody kinetics and durability to WT after vaccinated with inactivated COVID-19 vaccine. Then with the emergence of Δ and Omicron, we added the detections with limited reagents. The limit of detection we have added in all figures with your professional suggestions.

(2) It would be very useful to know what proportion of study participants were previously infected and if there are differences between the previously infected and uninfected individuals. This is clearly complicated by the fact that the vaccine is inactivated virus and therefore discrimination by the presence of anti-nucleocapsid antibodies will not work. However, any clinical record of infection could be used. If none, this should be stated.

Thanks for your great suggestion. It is very interesting to compare the antibody levels induced by inactivated SARS-CoV-2 in previously infected and uninfected individuals. The participants we recruited were not previous or later SARS-CoV-2 infected. Because our study was conducted from May to December 2021, when China had taken the strict measures to prevent and control the epidemic. Our organization (Zhejiang CDC) had the recorded information of individuals if they had been infected SARS-CoV-2. Meanwhile, the study regions had no large outbreaks and epidemics during that period. We have mortified the descriptions in the methods and results (line 116 and 117).

(3) Given the data, it is unclear how the authors can conclude on line 260 that: "protective capability of a primary vaccination series is enhanced following booster doses and optimization of the booster procedure" unless they are not talking about their study. The boost doesn't work as far as eliciting neutralizing antibodies against Omicron, which is what is circulating now, and this should be stated outright.

Thanks for your great suggestion. The description was not accurate and we have mortified in line 347-350:

“Despite the booster dose of inactivated vaccine can reverse the decrease of antibody levels to prime strain and heighten the cross immune response to Delta, it does not elicit potent neutralization against Omicron, which is circulating now. So, the optimization of the booster procedure such as heterologous boost immunization using viral-vector-, nucleic-acid-, and protein-based vaccines is necessary.”

Reviewer #2 (Recommendations for the authors):Line 115: 'critical' value. Did the authors mean threshold here?

Yes. Thanks for your great suggestion on improving the accessibility of our manuscript, we have corrected this word.

Lines 135-139 are unclear in terms of the exact correlations being tested for.

Thanks for your great suggestion, we have corrected the description in line 156-167:

“Correlation analysis showed poor correlation between NAb titers and anti-S/N IgM (R^2^ = 0.014, P = 0.243), NAb titers and anti-RBD IgM (R^2^ = 0.010, P = 0.322), NAb titers and anti-S/N IgG (R^2^ = 0.087, P = 0.003), neutralized fraction and anti-S/N IgM (R^2^ = 0.084, P = 0.003), neutralized fraction and anti-RBD IgM (R^2^ = 0.048, P = 0.028); whereas moderate correlations between NAb titers and anti-RBD IgG (R^2^ = 0.121, P < 0.001), NAb titer and neutralized fraction (R^2^ = 0.135, P < 0.001), neutralized fraction and anti-S/N IgG (R^2^ = 0.539, P < 0.001), neutralized fraction and anti-RBD IgG (R^2^ = 0.471, P < 0.001).”

Line 160 – may be better to use the term 'primary' course of vaccination.

Thanks for your great suggestion, we have corrected the descriptions in the full text.

Line 177: I would not use the word unprecedented. Rather could say they have been used widely.

Thanks for your great suggestion, we have corrected the descriptions.

Line 210-212: this does not make sense.

Thanks for your great suggestion, the sentences we described not clearly and have modified in line 250-252:

“We found that initial and booster-vaccine-elicited neutralizing antibody titers was weaker in older than younger adults, similar to the findings described in many other reports.”.

The references omit some of the key papers in the field. Where possible the authors should cite primary sources eg for the emergence of omicron Viana et al., Nature. For comparative biology of spike between omicron and δ, there are a number of possibilities including https://doi.org/10.1038/s41586-022-04462-1, https://www.tandfonline.com/doi/full/10.1080/22221751.2021.2023329, https://doi.org/10.1038/s41586-022-04474-x, and others. Some of the papers also report an increased breadth of neutralisation against omicron with booster doses that should be included in citations and compared to results observed in this study with boosting. https://doi.org/10.1038/s41586-021-04389-z.

Thanks for your great suggestion, we have modified the citations and added discussions about escapes against omicron in line 308-334.

In addition, the authors discuss the impact of age on response to vaccines but have not adequately cited papers on vaccine efficacy or in vitro neutralsiation / T cell responses in the elderly. This is clearly a key point for the Chinese population and needs further discussion, especially in light of recent developments in covid control policies.

Thanks for your great suggestion, we have added the discussions about the vaccine efficacy and vitro neutralization activity in the elderly in line 270-275.

The rationale for using an assay that detects N and S antibodies together without distinguishing them is unclear and should be addressed. In my view, this is a weakness.

We agree that the design is not perfect. There were not many commercial detection kits to choose that time. We additionally considered the inactivated vaccine had full virions, the detection of N and S antibodies together was in place of total viral antibodies, and we also detected the RBD antibodies.

Figure 4 should show data for Δ and Omicron. The data appear to be for WT only.

For Δ and Omicron, we only detected the samples of 28 days after the second and third dose (V2-1m and V3-1m). Due to the weaker neutralized fraction to Delta and Omicron, we did note detected samples of other groups. The results showed in Figure 5.

Table 1: Please define how the positive rate was determined.

According to the manufacturer of kits, the concentration anti-S/N antibodies≥10.0 AU/mL (IgG) and ≥1.0 AU/mL (IgM) are considered positive (or reactive); for anti-RBD antibodies, the positive was defined as concentration≥1.0 AU/mL. The positive rate was percent of numbers of positive samples/ total detected samples. We have added the description in the table legends. Thanks for your great suggestion.

Table 3: The 'positive rate' should be defined. Is it a GMT of >20 for example?

The seropositivity rate is when positive titer of NT50 is above 1:4 and descripted in the methods. Thanks for your great suggestion, we have added the description in the table legends.

Table S2: please define P1 and P2 in the table legend. There should be 95% CI or sd around the inhibition rate %.

Thanks for your great suggestion. We have added the description in the table S2 legends. P1 and P2 means serum sample of person1 and person2. The samples were conducted with the serial dilution from 1:100 and tested with pseudovirus-based neutralization. We only detected with 1 complex well that 95% CI or sd around the inhibition rate % were valueless.

Table S3: The 'positive rate' should be defined.

We have added the description in the table S3 legends. Thank you very much for your valuable revision suggestions.

[Editors’ note: further revisions were suggested prior to acceptance, as described below.]

The authors partially completed the referee corrections, and this paper does not require an additional round of review. However, the presentation of the work should be corrected before publication.(1) The most significant result of the study – that the inactivated vaccine booster does not elicit potent neutralizing antibodies against Omicron, should be stated in the abstract.

Thanks for your great suggestion. We have mortified the description in line 44-45, 347-350.

(2) Title: "Evaluation of antibody kinetics and durability in health subjects…" should be changed to "Evaluation of antibody kinetics and durability in healthy…". Also, the term "subjects" is outdated, and the authors should consider using "individuals" or "participants".

Thanks for your great suggestion. We have corrected the title to “Evaluation of antibody kinetics and durability in health individuals vaccinated with inactivated COVID-19 vaccine (CoronaVac): A cross-sectional and cohort study in Zhejiang, China**”**

(3) Figure 2: that cross-sectional and not cohort (if that is the case) study participants were analyzed should be stated in the legend. If the participants at each time point are not the same, there should be no connecting lines between points (e.g., a bar graph should be used).

Thanks for your great suggestion on improving the accessibility of our manuscript. We have corrected in the Figure 2.

(4) Figure 3A-B: labels on graph missing "2" (e.g., V-1m instead of V2-1m).

Thanks for your great suggestion. It was my mistake and we have corrected in the Figure 3A-B.